# Non-Destructive Techniques for the Analysis and Evaluation of Meat Quality and Safety: A Review

**DOI:** 10.3390/foods11223713

**Published:** 2022-11-18

**Authors:** Xiaohong Wu, Xinyue Liang, Yixuan Wang, Bin Wu, Jun Sun

**Affiliations:** 1School of Electrical and Information Engineering, Jiangsu University, Zhenjiang 212013, China; 2High-Tech Key Laboratory of Agricultural Equipment and Intelligence of Jiangsu Province, Jiangsu University, Zhenjiang 212013, China; 3Department of Information Engineering, Chuzhou Polytechnic, Chuzhou 239000, China

**Keywords:** meat quality, non-destructive detection, near-infrared spectroscopy, Raman spectroscopy, hyperspectral imaging

## Abstract

With the continuous development of economy and the change in consumption concept, the demand for meat, a nutritious food, has been dramatically increasing. Meat quality is tightly related to human life and health, and it is commonly measured by sensory attribute, chemical composition, physical and chemical property, nutritional value, and safety quality. This paper surveys four types of emerging non-destructive detection techniques for meat quality estimation, including spectroscopic technique, imaging technique, machine vision, and electronic nose. The theoretical basis and applications of each technique are summarized, and their characteristics and specific application scope are compared horizontally, and the possible development direction is discussed. This review clearly shows that non-destructive detection has the advantages of fast, accurate, and non-invasive, and it is the current research hotspot on meat quality evaluation. In the future, how to integrate a variety of non-destructive detection techniques to achieve comprehensive analysis and assessment of meat quality and safety will be a mainstream trend.

## 1. Introduction

Meat and meat products are major sources of high-quality protein and they are rich in vitamins and minerals. Beef, pork, mutton, and other raw meat can replenish the microelement, such as iron lacking in human body and they are an indispensable food to strengthen the physical function and promote metabolism [1,2]. In recent years, with the continuous improvement in consumption level, people gradually turn to the pursuit of quality of life. To maintain life and health and improve the irrational diet structure, the demand for meat, a type of nutrient-rich product, is increasing day by day [3]. The consumption of meat per person is expected to increase to 35.5 kg by 2024 according to the Organization for Economic Co-operation and Development (OECD).

Meat is a diverse product. Internal factors of livestock and poultry themselves, or external conditions, such as rearing environment, slaughter time, and storage temperature, will have a great impact on the quality of meat [4]. Once exposed to light, dust, and microorganisms during transportation and sales, they will take on an unappetizing appearance. The quality of meat is directly related to the survival and development of human beings, and it is the most critical aspect for consumers to consider when purchasing meat. Some consumers are even willing to pay a higher price to guarantee the quality of meat [5]. However, in recent years, many unscrupulous merchants have tended to take risks to seize the market and obtain huge profits, resulting in various quality problems. Government departments and the food industries should pay close attention to the safety and quality of meat to safeguard the legitimate rights and interests of consumers [6]. Therefore, how to test and estimate the quality of meat has become the top priority of research.

Sensory attribute, chemical composition, physical and chemical property, nutritional value, and safety quality are five important standards commonly used [7]. Standards of sensory attribute include color, smell, taste, and texture. Color, a basic attribute of food, is the external embodiment of physical and chemical properties, which can directly reflect the freshness of meat and meat products. Meat mainly contains six substances, namely water, protein, fat, vitamins, minerals, and carbohydrates, which are closely related to its nutritional value. Water is the most abundant ingredient in meat and its specific content and distribution, especially water-holding capacity (WHC), which will affect the taste of meat. Tenderness and juiciness depend on the composition of the fat, determining the purchase intention of customers. Among the physical and chemical properties, pH value can be used to grade the quality of meat into RFN (reddish-pink, firm, non-exudative), PSE (pale, soft, exudative), and DFD (dark, firm, dry). Safety qualities such as freshness, authenticity, and adulteration are tightly linked to human health.

Traditional discrimination methods of meat are broadly divided into two categories: Subjective and objective, including sensory assessment, microbial detection, physical and chemical experiments, etc. Sensory evaluation, integrating senses of sight, smell, taste, and touch, is an artificial measurement. It depends on the experience and practice of inspectors, and the results are often difficult to be quantified. Objective evaluation refers to ascertaining the physical and chemical properties of meat by means of a scientific experiment with the aid of instruments. There is a certain improvement in the speed and accuracy of results while it will cause irreversible damage to products [8]. To improve the detection efficiency and reduce the loss of products, modern techniques are developing rapidly in the direction of rapid, accurate, and non-destructive detection.

Non-destructive detection technique (NDDT) is a burgeoning comprehensive subject based on physics, electronics, computer science, artificial intelligence (AI), etc. It can describe the internal structure and external properties of a substance by detecting the optical, acoustic, and electromagnetic characteristics without destroying its original state [9]. NDDT is a promising method in the field of meat quality inspection due to its virtues of rapidness, real-time performance, accuracy, and non-destructive detection. AI has developed into one of the most revolutionary technologies in the 21st century. It provides technical support for online meat grading and evaluation, and it is expected to bring about an unprecedented opportunity for the non-destructive detection of food and agricultural products [10].

In this paper, four types of non-destructive detection techniques commonly used to estimate the quality of meat and meat products in recent years have been reviewed in detail, including spectroscopic techniques (near-infrared spectroscopy, Raman spectroscopy, and terahertz spectroscopy), imaging techniques (hyperspectral imaging, X-ray imaging, and thermal imaging), machine vision, and electronic nose. The review discusses the theoretical basis and current applications of these emerging techniques, as well as the characteristics and challenges faced by them, and finally presents the outlook for future development directions. The searches on the research articles were carried out using several databases, such as Web of Science and Science Direct. The keywords focused on different types of meat and meat products in combination with non-destructive detection techniques.

## 2. Spectroscopic Techniques

Spectrum can be divided according to the range of wavelength, which is successively from small to large: γ-rays, X-rays, ultraviolet, visible light, infrared, microwave, and radio waves [11]. Spectroscopy is not only an interdisciplinary subject combining physics and chemistry, but also an important measurement for qualitative and quantitative analysis of meat and meat products. Spectroscopic techniques, including near-infrared spectroscopy (NIRS), Raman spectroscopy (RS), and terahertz (THz) spectroscopy, have been used in food, communication, and health care as well as other areas, and are often combined with various approaches in practical applications. Multivariate statistical analysis (MSA) methods include multiple regression analysis (MRA), cluster analysis (CA), and discriminant analysis (DA), and have been greatly used to process the spectral data. Moreover, stoichiometry and computer technology have made great contributions to the development of spectroscopy [12].

### 2.1. Near-Infrared Spectroscopy

Near-infrared spectrum is an electromagnetic wave between the visible spectrum and mid-infrared spectrum, with a wavelength range of 800 to 2500 nm. Different organic matter often contains different hydrogen-containing groups, such as O-H bonds in water, N-H bonds in protein, and C-H bonds in fat. When exposed to infrared spectrum, they are excited into resonance and can absorb the energy. Molecules absorb each wavelength at a specific rate, creating peaks and troughs in the spectrum. The principle of near-infrared spectroscopy is to measure the frequency doubling and frequency merging absorption of these hydrogen-containing groups, and then conduct qualitative or quantitative analysis of the components according to the position and intensity of the absorption spectrum [13].

Sample preparation indicates the selection of a batch of representative meat and meat products with different parts, varieties or origins and its process will partly affect the reliability of results. In general, NIRS tends to be more accurate in predicting the chemical composition of chopped tissue than intact meat samples due to the homogeneity of ground products [14]. The energy absorbed will be lower after the crushing treatment, leading to a higher reflectance, which is easier to be measured. Therefore, meat and meat products are often physically segmented before subsequent data acquisition and the finer the grinding, the better the results [15].

Frequently-used devices, to scan the samples and collect their spectral data, include Fourier near-infrared spectrometer, portable near-infrared spectrometer, etc. At present, Denmark and Germany have developed equipment that can be applied to online detection in industrial production. During the procedure of data preprocessing, different methods should be appropriately selected on the basis of persisting issues. Smoothing is mainly applied in eliminating the interference of high frequency noise [16]. Utilizing first and second derivatives can remove the influence of systematic background, such as base-line drift. Additionally, the scattering correction, including multiplicative scattering correction (MSC), orthogonal signal correction (OSC), etc. can effectively reduce the adverse effects of scattering on the model [17]. Moreover, averaging and centering are commonly used spectral preprocessing methods.

When quantitatively analyzing the physical and chemical parameters of samples, the data obtained based on standard methods are the theoretical limit of using mathematical models. Acid value (AV) is defined as milligrams of potassium hydroxide needed to neutralize the free fatty acids in one gram of fat, and it is an important index to measure the degree of hydrolysis of fat [18]. Kjeldahl method can detect the content of total volatile basic nitrogen (TVB-N), which has been widely used to estimate the crude protein content in beef. Soxhlet extraction and drying method are always applied for the determination of the fat and moisture content [19,20].

The most critical and complex procedure of NIRS is to establish the mathematical relationship between spectral information and physicochemical data, including analytical method, prediction equation, etc. The selection of spectral region is generally based on the specific characteristics of samples. In general, the system will contain more information if the spectral range is sufficiently comprehensive. Enlarging the counts of scans and expanding the sampling interval are two effective means to improve the prediction accuracy in daily application [21,22]. However, the error in measurement will increase in the meantime with the extension of data points, thus the choice of spectral region is necessary to avoid the part of small information and large distortion. To solve the influence caused by overlapping spectral peaks and the interference based on a complex background, all the information of the selected spectral region must be applied in the modeling process. The data processing algorithms commonly used include principal component analysis (PCA), linear discriminant analysis (LDA), support vector machine (SVM), partial least squares regression (PLSR), etc.

Cross-validation is the most common method to evaluate the prediction accuracy of mathematical models. It takes out a fixed number of samples each time, using the remaining sections to establish a model and detect the samples chosen before, and finally it repeats the above operations until all of them have been tested. When the correlation coefficient approaches 1 and the corrected standard deviation is close to the measured standard deviation, the performance of the system is best. In addition, external validation has been applied in researches on meat and meat products, such as pork and pork sausages [23,24]. After evaluating the system performance (accuracy, stability, etc.) and further improving the model that has been built, the quality of meat can be directly calculated by collecting the spectra of unknown samples and then using this model.

Compared with traditional chemical analysis methods, NIRS has the advantages of rapidness, non-destructive detection, and low cost, and has been widely used for chemical composition analysis, edible quality evaluation, and adulteration identification. Chemical composition is the basic property of food, and the current researches on meat and meat products have focused on the content of moisture, crude protein (CP), and intramuscular fat (IMF). As early as the 1960s, there was a study on the determination of water and fat content in meat emulsion using direct spectrophotometric techniques [25]. Thereafter, with the introduction of chemometrics, modern optics, and computer data processing, NIRS has developed rapidly. Prieto et al. successfully used near-infrared reflectance spectroscopy to estimate several chemical parameters of oxen meat, indicating that NIRS was an useful tool to analyze the composition of beef [26]. Other researches have confirmed the ability of NIRS in the field of meat products. For example, Gaitan-Jurado et al. were devoted to evaluating the content of fat, protein, and moisture in pork dry-cured sausages using the diode array instrument [27]. In addition, there are abundant studies on the composition analysis of mutton, breast meat, etc. [28,29].

Edible quality mainly contains tenderness, marbling, color, etc. Tenderness is a specific characteristic to describe the texture of meat, referring to the softness and fragility of the mouthfeel when tasting. At present, Warner-Bratzler shear force (WBSF) is widely used as a criterion to determine meat tenderness. Barlocco et al. predicted the IMF and moisture content and WBSF in intact and homogenized pork muscles based on PLS models, respectively, suggesting the potential of NIRS to estimate the content of IMF and moisture in homogenized tissues and WBSF in intact samples [30]. Byrne et al. used PCA to discuss the correlation between NIRS and the selected quality attributes (tenderness, marbling, and flavor) of raw beef [31]. Cozzolino et al. identified the color of pork muscle using visible and near-infrared reflectance spectroscopy combined with modified partial least squares analysis [32].

Adulteration often has the characteristics of low investment and high return. It is common to replace high-quality meat with raw meat of other varieties and origins. In addition, soybean and other vegetable proteins are often added in meat products to achieve the purpose of reducing costs. Meanwhile, it is hard to tell the difference between real meat and fake meat owing to the complex composition and diverse adulteration methods. Therefore, how to accurately detect the safety quality of meat is of great significance. Kuswandi et al. established a PLS and LDA model to determine pork adulteration in beef meatball based on chemometrics, and the results were in good agreement with the immunochromatographic method [33]. Alamprese et al. aimed at evaluating the ability of visible, near-infrared, and mid-infrared spectroscopy for identification and quantification of turkey meat adulteration in minced beef by different multivariate regression and class-modeling strategies [34]. Additionally, NIRS could be used for the detection of adulteration in beef hamburgers, with an accuracy of up to 92.7% [35].

However, this technique still has some limitations, such as the demand for a large number of physical and chemical experiments before modeling. Machine vision is the most widely used method when detecting meat color, and the accuracy of using electro-magnetic characteristics is often higher than NIRS in the aspect of freshness. Therefore, NIRS should be combined with other non-destructive detection techniques in the comprehensive evaluation of meat quality to increase the detection efficiency and economic benefits. Moreover, due to the great influence of external factors on the spectrum, how to effectively extract spectral information and increase the signal-to-noise ratio is an important direction in the future research.

### 2.2. Raman Spectroscopy

The scattering is classified as inelastic scattering or Raman scattering if the incident photons exchange energy with molecules during the collision. Raman spectrum is a type of molecular vibration spectrum based on Raman scattering effect [36]. It detects the inelastic scattering spectrum generated by the interaction between lasers and molecules to obtain the vibration or rotation information of molecules and plays a complementary role with the near-infrared absorption spectrum [37]. In fact, each molecule has its own unique Raman spectrum signal. Therefore, once the special molecular information in the samples has been extracted, the qualitative judgment of the material structure can be successfully realized. In the field of meat detection, chemometrics is often used to extract features from samples. By analyzing the representative information of meat samples, the relationship between the molecular structure and various free radical groups can be possessed to further evaluate the quality of meat.

Raman spectroscopy, a green detection technique, does not require pretreatment of samples and has certain advantages in generating information. In general, only a small number of samples are needed to obtain the key characteristics of compounds in food. In recent years, Raman spectroscopy has been increasingly applied in the prediction of meat dietary quality traits. Schmidt et al. conducted investigations on the shear force and cooking loss of 140 raw sheep meat samples from two different origins with a prototype handheld Raman system. They used PLSR algorithm to correlate the Raman data with quality traits and the results exhibited the usefulness of Raman spectroscopy to estimate the tenderness and cooking loss of meat [38]. Similarly, Fouler et al. focused on the juiciness and tenderness of 45 beef loins using a 671 nm handheld Raman spectroscopic device, and the correlations between predicted and observed values were 0.42 and 0.47, respectively [39]. Moreover, Raman spectroscopy has considerable potential to determine the quality attributes of cooked meat. Beattie et al. selected 52 cooked beef samples and successfully measured their texture and tenderness [40].

In the analysis of chemical components, Raman spectroscopy is often used to study the changes in chemical bonds in substances, especially for the determination of water, protein, and lipid structures in meat. Pedersen et al. combined infrared absorption spectrum with Raman scattering spectrum to reflect the moisture content of pork in a short period after slaughter, indicating the feasibility of Raman spectroscopy to determine the structure of water in biological macromolecules [41]. The Raman bands of proteins are usually assigned based on the model compounds in meat, such as amino acids and short peptides. Herrero used Raman spectroscopy as a powerful and non-invasive method to provide the information on the secondary structure of muscle proteins, and the results demonstrated its possibilities to predict the functional properties and sensory attributes of protein in intact muscle and muscle food [42]. Moreover, Shao et al. investigated the changes in structure, texture, water- and fat-binding capacity of raw and heated meat batters added with different lipids. The results showed that both the preparation process and thermal treatments will cause great damages to the properties of meat batters [43].

Applications of Raman spectroscopy in safety quality evaluation revolve around spoilage, adulteration, etc. Based on the usage of machine learning in combination with evolutionary computing methods, Argyri et al. compared Raman spectroscopy with Fourier transform infrared (FT-IR) spectroscopy on the prediction of meat spoilage. It turned out that both Raman spectroscopy and FT-IR spectroscopy reliably assessed the spoilage of meat [44]. Boyaci et al. extracted the pure fat samples from 49 beef and horsemeat samples and used PCA to perform the data mining process of Raman spectra, providing an accurate and fast method to identify the beef adulteration with horsemeat based on Raman spectroscopy and chemometrics [45]. Boar taint is a disgusting odor that results from the accumulation of androstenone and skatole, especially in the fat tissue of non-castrated male pigs, and using a portable Raman device is a feasible approach to detect and classify different types of boar taint [46].

In recent years, with the progress of laser technology and nanotechnology, Raman spectroscopy has been comprehensively applied in the field of meat quality evaluation. It is a direct and non-destructive technique compared with traditional detection methods. However, in view of the heterogeneity and complexity of meat and meat products, future researches need to determine the repeatability and robustness of models which have been established on a larger independent dataset. In addition, Raman spectroscopy is still a promising direction for scientific studies and industrial applications to explore the law of quality change in meat processing.

### 2.3. Terahertz Spectroscopy

Terahertz (THz) wave belongs to the far-infrared band and its frequency is in the range of 0.1 to 10 THz, which is located between millimeter wave and infrared spectrum [47]. As a promising direction in the field of spectroscopic technique, THz spectroscopy benefits from its unique radiation band. It uses THz rays to irradiate the measured objects and can be available to obtain the information through the transmission or reflection of samples, which has the advantages of high signal-to-noise ratio, wide dynamic range, etc.

According to the type of sources, THz imaging systems can be divided into two modes: Pulse and continuous. The former has the characteristics similar to terahertz time domain spectroscopy (THz-TDS) and can perform functional imaging of objects to obtain the refractive index distribution inside the substance. Although the latter has some advantages in the complexity of data and system, it cannot reflect the comprehensive information of the samples [48,49]. Therefore, it is necessary to combine the characteristics of the two methods and then select the imaging system according to the specific application. Terahertz spectroscopy is able to determine the appearance and internal components of targets by collecting their time domain and frequency domain information in THz band and has been deeply used for the detection of meat quality.

For the sake of financial interests, some enterprises cheat on the quality of meat, such as selling defective merchandise and putting additives into the raw meat. THz spectroscopy can identify the meat from different tissues, varieties or even different brands of the same variety, and this provides the theoretical foundation and experimental basis for the authenticity and adulteration of meat in practical application. It is widely known that many materials have characteristic absorptions in terahertz range. THz-TDS can be used to ascertain the absorption and refractive index of tert-butylhydroquinone (a food additive in McDonald’s chicken products). The results indicate that it is a potential method to detect the food additives in meat products [50]. In addition, THz spectroscopic imaging has the ability to distinguish the foreign materials in food and has effectively located metal contaminations in sausages with complicated compositions using PCA combined with discriminant analysis methods [51].

During the deterioration process, the absorption coefficients of pork tissue are often different in the time domain and frequency domain. Terahertz spectroscopy possesses the fingerprint characteristics of many biomolecules and covers a large number of material vibration models. It can be used to respectively detect the freshness of preserved meat and spoiled meat, and the results indicate that the moisture content is closely related to meat quality, which provides a reference for the application of this technique in meat freshness identification [52]. A study on the spoilage of salmon using THz-TDS and electrochemical impedance spectroscopy confirms that Terahertz spectroscopy is a non-invasive and non-destructive method to monitor the quality of fish [53]. Terahertz spectroscopy presents a multifaceted capability to observe the low-energy response of macromolecules, cells, and tissues, and to determine the biophysical effects of terahertz wave. In the detection of meat products, lean meat always absorbs terahertz radiation, while fat meat is almost transparent to the radiation, which can be used to detect the proportion of fat and lean [54].

Terahertz wave has a strong penetration to most dielectric materials and non-polar substances, and its energy is lower than the bond energy of various chemical bonds, thus it will not cause harmful ionization reactions. Therefore, THz spectroscopy has been widely used in semiconductor materials, broadband communication, microwave orientation, etc. In the field of meat and meat product detection, there are many potential problems in the application and research of this technique. First, its theoretical research is still in the early stage. For instance, the reason for the signal generation of matter in the terahertz band range is relatively complex and needs to be further studied. Second, water has a strong absorption of terahertz radiation and the meat samples always contain a large amount of water, which will affect the signal-to-noise ratio in the testing process. Finally, the experimental platform for transmitting and receiving terahertz waves is also necessary to be improved. Table 1 shows the extensive applications of spectroscopic techniques in the field of rapid non-destructive detection for meat quality in the past years.

## 3. Imaging Techniques

In recent years, with the iterative update of camera performance and the continuous improvement in computer hardware processing capacity, imaging techniques gradually present the potential in the field of non-destructive detection and have been widely used in aerospace, agriculture, food, and other areas [83]. Imaging is a non-contact method using optical principles, and it can provide the spatial information of objects related to their chemical properties and sensory attributes, which is crucial for measuring the external characteristics of food. Emerging imaging techniques commonly used for meat quality and safety evaluation include hyperspectral imaging (HSI), X-ray imaging, and thermal imaging (TI), which will be separately described in the following sections.

### 3.1. Hyperspectral Imaging

Spectroscopy aims to obtain spectral information by measuring the optical properties of food [84]. Additionally, the spectral information can reflect the chemical composition of meat, which is of great significance for estimating its internal quality characteristics. However, there are some limitations in the application scope of spectral technology. Computer vision is the most commonly used method when describing the distribution in space, and the acquired image information can be used to detect the external attributes of meat products. Hyperspectral image uses the voxels to characterize spectral information and can describe the two-dimensional spatial information of objects, which is a comprehensive hyperspectral cube in three-dimensional space. The approaches to acquiring hyperspectral images include point scanning, line scanning, area scanning, and single shot scanning [85]. Among them, line scanning is the most popular scanning method concerning food quality and safety assessment [86]. Hyperspectral imaging (HSI) is a new generation of photoelectric detection and fusion technology and has many characteristics, such as high spectral resolution, continuous multi-band, and atlas integration [87,88]. Therefore, HSI, as the name implies, integrates the advantages of traditional spectroscopy and imaging technology, which can provide spectral information and image information at the same time, providing a feasible method for qualitative discrimination and quantitative analysis of comprehensive meat indicators.

The applications of HSI in meat quality evaluation mainly focus on sensory attributes and chemical components. Tenderness is one of the most critical characteristics to describe meat palatability. Cluff et al. developed a non-destructive model to classify the tenderness of cooked-beef based on hyperspectral optical scattering imaging, and the accuracies were 83.3% and 75.0% for tough and tender samples, respectively [89]. Similarly, Wu et al. used hyperspectral scattering techniques to predict the color, pH value, and tenderness of beef, with a correlation coefficient of 0.86 for WBSF [90]. Additionally, they obtained a better correlation coefficient of 0.91 in further research [91]. Some researches concentrate on the other sensory characteristics of meat. Aredo et al. scanned 58 longissimus dorsi muscle samples by the HSI reflectance mode and established a model using PLSR algorithm to predict the marbling of beef, which turned out to be a high correlation coefficient of 95% in the prediction [92]. Moreover, Kamruzzaman et al. used a visible HSI system in line scanning mode to measure the color parameters of red meat and the outcome showed that HSI imaging was a potential technique to evaluate meat color [93].

In the field of chemical components, applications of HSI revolve on the detection of moisture, protein, and fat content. Water is a main component of meat and can be precisely determined by hyperspectral imaging. The HSI systems with a wavelength ranging from 900 to 1700 nm have been used for the detection of moisture content in raw meat such as pork, beef, lamb, etc. [94,95,96]. Furthermore, HSI facilitates the estimation of water in meat products. On the basis of near-infrared hyperspectral imaging in combination with chemometrics, Achata et al. developed a novel prediction model to determine the content of water in beef jerky samples, indicating the satisfaction of NIR-HIS to evaluate drying behavior in cooked meat [97]. Talens et al. applied multivariate analysis methods (PLSR and PLS-DA) to predict the contents of water and protein in Spanish cooked hams and successfully graded the examined samples into different quality categories [98]. Moreover, the study on the content and distribution of intramuscular fat is an important direction of hyperspectral imaging. Liu et al. successively employed stepwise procedures and PLS to predict the IMF content of pork related to juiciness, tenderness, and taste, with the adjusted R^2^ of 0.92 and 0.93, respectively [99]. Lohumi et al. visualized the concentration of IMF in beef in the spectral range of 400 to 1000 nm, providing a fast and accurate method for the determination of IMF distribution [100].

HSI is a potential non-destructive detection technique in the field of food quality and safety assessment. However, due to the fact that hyperspectral images cover both spectral and spatial information of the objects, HSI often has large and complex amount of data and cannot be directly applied for online meat quality detection. Therefore, it is indispensable to select suitable analysis algorithms to process the data in advance, which can effectively save computation time. Additionally, choosing the most relevant wavelength is instrumental in eliminating the variability of spectral data. Moreover, the rationality of spectral regions has a great influence on the development of HSI. Utilizing different image processing methods, such as artificial neural network (ANN) in selected wavebands, the region of interest can be finally determined [101]. At present, HSI technique has been widely used in the detection of defective fruits, damaged mushrooms, teas, etc. In the future, HSI will develop in the direction of high speed and low cost, and the emergence of a real-time food monitoring system to satisfy the requirement of modern industry is anticipated [102].

### 3.2. X-ray Imaging

The X-ray imaging system often consists of X-ray generator, linear array detector, image acquisition card, and display equipment [103]. X-ray has the characteristics of penetration, diffraction, and fluorescence excitation. The transmission and tomography images of samples can be obtained by capturing the penetration characteristics of X-rays. Computed tomography (CT) is one of the most popular X-ray imaging techniques at present. It is often used in the quality and safety inspection of food, especially providing a great possibility for research on the structure and components of meat. CT has less damage to the objects and can be used to investigate exactly the same samples before and after heat treatment. Miklos et al. used X-ray tomography combined with 3D image segmentation to quantitively study the heat induced structural changes in meat, especially the internal components of samples including water, fat, connective tissue, and myofibrils [104]. Furnols et al. estimated the lean meat percentage (LMP) in pig carcasses using PLSR algorithm to obtain a calibration equation for the computed tomography scans [105].

In general, the freshness of meat will be greatly reduced after the freezing treatment. Many unscrupulous traders cut costs by replacing raw meat with frozen meat. Kobayashi et al. conducted a comparative study of tuna meat before and after freezing by computed tomography, and the results showed that uneven ice crystal structures would form in the cells of frozen tuna meat, which provided an idea for distinguishing frozen meat [106]. Diffraction-enhanced imaging is a new radiographic imaging modality used for NDDT and X-ray computed tomography [107]. It is an important method to distinguish fresh meat from frozen meat. The diffraction intensity is directly proportional to the freshness of meat and the quality of meat can be judged by the position of wave peaks in the diffraction pattern [108]. In addition, the characteristics of meat change greatly after being processed into meat products and the X-ray imaging technique usually has accurate prediction results when detecting the microstructure characteristics of meat products, such as fried chicken nuggets [109].

X-ray imaging plays an important role in the field of food safety detection. It has been commonly used to determine the foreign materials in meat and meat products to avoid food safety accidents in daily life. Tao et al. developed a new image detection algorithm to detect the bone fragments in meat, and it was feasible even in the samples with uneven thickness [110]. However, when the density of foreign materials is close to water, the detection results of this technique are often not very good. In addition, X-ray imaging can be used to qualitatively analyze the mildew of food. Through parameter analysis and processing of the mildew image, effective detection of the mildew area can be realized.

### 3.3. Thermal Imaging

Thermal imaging is a type of non-destructive and non-contact temperature sensing technique. It is one of the most commonly used food contaminant detection methods. Thermal imaging technique mainly collects thermal infrared band light to detect the thermal radiation [111]. An infrared thermal imaging system consists of a thermal camera equipped with an infrared detector, a signal processing unit, and an image acquisition system. Infrared detectors absorb infrared energy emitted by objects and convert it into electrical signals. Then, the pulses are sent to the signal processing unit, which transforms the information into thermal images. The use of thermal imaging method avoids the risk of contamination caused by contact measurement and can quickly obtain the temperature value and its distribution at a specific point. According to these characteristics, it can be non-invasively applied to food detection.

Thermal imaging can generate images by collecting thermal radiation emitted by objects without interference from the external environment. The infrared thermal imaging temperature measurement has been developed greatly due to its advantages of accurate results and non-contact process. Infrared thermal imaging is a popular thermal imaging technique. Traffano-Schiffo et al. used infrared thermal imaging to obtain changes in meat surface temperature during the drying operation [112]. Kor et al. compared two different methods, contact (thermocouples) and non-contact (thermal imaging), to measure the temperature of semi-cooked cylindrical minced meat products during cooking. The results showed that the thermal imaging non-contact temperature measurement method is faster and can greatly reduce the errors of point measurement [113].

Driven by market interests, some illegal merchants mix low-price meat with high-price meat to obtain higher profits. Zheng et al. collected 35 samples of pure mutton, 35 samples of pure pork, and 175 samples of adulterated mutton in the experiment. The combination of thermal imaging and convolutional neural network (CNN) achieved great results in qualitative classification of different samples and quantitative prediction of the adulteration ratio, and the accuracy of validation set and test set were 99.97% and 99.99%, respectively [114]. Moreover, thermal imaging can be used to predict the temperature of chicken after cooking combined with multi-layer neural networks, demonstrating its potential in estimating the doneness of chicken [115].

Recent researches on the evaluation of pork and beef products using the thermal imaging technique mainly focus on the alive samples. Cuthbertson et al. investigated the real-time quality and physiological response of cattle exposed to transport using infrared thermography [116]. Newborn piglets are susceptible to the environmental temperature and the cold condition is one of the most dangerous stressors encountered. Tabuaciri et al. intended to treat the thermography as an early diagnostic tool to distinguish hypothermia piglets [117].

Thermal imaging is a promising technology, which can be applied in a wide range of fields. It has the advantages of long distance, strong penetration, resistance to strong light interference, and can adapt to the working environment at night and under harsh conditions. However, thermal imaging has limitations. It records the heat distribution by measuring the infrared radiation emitted from the surface of objects to map the temperature. Therefore, it is sensitive to thermal interference in the environment, and the results are often uncertain when detecting targets with unstable temperature. In addition, the choice of technology in measurement of temperature has a great impact on the prediction accuracy and the contact method generally shows more reliable results. Thermal imaging is developing toward low cost and high accuracy. Table 2 lists the typical applications of imaging techniques as non-destructive detection methods for meat quality attributes during the recent years.

## 4. Machine Vision

Machine vision technology improves the quality of vision through electronic perception and image investigation. Image sensors acquire target images, which are analyzed and converted into digital information by computer technology, and then identify and detect target objects [144]. It is a comprehensive, efficient, fast, and non-destructive technique that objectively obtains accurate, reliable, and repeatable data. Moreover, it can replicate and replaces human vision and perception of images. However, this technique is limited to identifying external factors, such as color and size when analyzing digital images. It requires artificial illumination in dimly lit scenes [145].

Machine vision system consists of image capture equipment, light sources, and computer hardware and software. The devices used for image capture can be digital cameras, ultrasound scanners, computed tomography scanners, etc., enabling the user to obtain information about the external and internal characteristics of the object under test [146]. Illumination level is one of the important influencing factors for machine vision systems. The appearance of the object can be changed to some extent by external illumination to make the features of the measured part clearer and reduce reflections, shadows, and noise [147]. Moreover, image processing is the core process of machine vision systems. Low-level processing mainly refers to image pre-processing (noise reduction, gray level, geometry, and bokeh correction). Mid-level processing includes image segmentation, display, and description. High-level processing includes image diagnosis and annotation, aiming at transforming the extracted data into valid information [145].

### 4.1. Image Acquisition Method

Machine vision techniques are now widely used in feature extraction and recognition of meat-related images [144]. The first step of a machine vision system is illumination and image acquisition, image acquisition by digital camera or smartphone shooting, ultrasound imaging, nuclear magnetic resonance (NMR), computed tomography (CT), near-infrared spectral imaging (NIR), hyperspectral imaging (HS), etc.

#### 4.1.1. Camera or Smartphone Shot

Machine vision technique was investigated to detect defects in pork loin longus muscle in an industrial setting [149]. Computer vision system (CVS) could detect PSE and DFD, but could not distinguish between RSE and RFN [148]. Moreover, it shows that CVS can analyze the color of the entire meat surface in a non-invasive manner. Huang et al. used two 18 W illumination lamps to generate light diffusion and a high-performance charge-coupled device (CCD) camera to acquire images to determine the TVB-N content in pork [149]. Huang et al. applied fluorescent lights to provide uniform illumination, LED lights as supplemental illumination, and a CCD camera with an 8 mm zoom lens to acquire images to assess fish freshness using CV and NIR. BP-ANN achieved 90% prediction success in CV, and 80% in NIR, and after data fusion, the prediction success reached 93.33% [150]. The CVS can reproduce colors very similar to the true colors and does not produce large color differences due to translucency or inhomogeneity of the meat matrix, making it easier to preserve sample images [151]. Determination of the shelf life of fish by CVS analysis is fast, non-destructive, and correlates well with chemical and sensory results [152]. Moreover, it was demonstrated that CVS was able to predict the TVB-N, TVC, and TBA of both pupils [153].

#### 4.1.2. Ultrasound Imaging

Ultrasound imaging is a cost-effective technique to obtain images of the interior of the sample under test. Typically, there are two modes of ultrasound imaging: Amplitude modulation and brightness modulation, with brightness modulation being more widely used [154]. Fukuda et al. accurately estimated the number of beef marble standards (BMS) using ultrasound echo imaging [155]. Utilizing independent component analysis (ICA), the resulting correlation coefficient between actual and estimated values was 0.7, which was higher than PCA (r = 0.6) [155]. Brethour acquired images using an Aloka 210 ultrasound system equipped with a 3.5-MHz universal transducer array to obtain tomograms of the longest lateral spinal muscles [156]. Aass et al. predicted the fat content in lean cattle by a scanner equipped with QUIP software and an ASP-18 transducer for image acquisition [157].

#### 4.1.3. Nuclear Magnetic Resonance

NMRI has different features in the electromagnetic spectrum to assess meat quality and safety. Different biochemical properties of the subject result in different absorption and emission of energy in the electromagnetic spectrum [154]. Avila et al. studied the distribution of several texture features inside meat products by MRI, and the parameters obtained by traditional destructive techniques are highly correlated with the data obtained by texture analysis based on volumetric information, while the CVS by 3D algorithm allows for the attainment of quality parameters in a non-destructive way [158]. A horizontal 4.7T NMRI system was prepared for image acquisition to achieve in situ dynamic imaging of the connectivity network during meat cooking by NMRI, allowing for the simultaneous monitoring of local deformation and changes in water transfer, muscle structure, and thermal history [159]. Kremer et al. evaluated whether magnetic resonance imaging (MRI) can reliably analyze components in pigs [160]. Compared with other CVS imaging methods, MRI is more complex and time-consuming and requires a high level of imaging samples, but its non-invasive nature enables in vivo imaging, which is more convenient for follow-up studies of samples.

#### 4.1.4. Computed Tomography

X-ray computed tomography uses X-rays to create a tomographic image of the scanned sample. When the tissue in the sample under test attenuates the X-rays, a thin cross-sectional image of the sample is obtained [154]. The new muscle indices were developed for the hind legs and lumbar region of lambs by CT assessment of muscle mass and bone size [161]. An X-ray computed tomography method was improved to automatically detect fish bones in fish fillets, validated using salmon and trout, achieving very high classification rates in quality control, with cross-validation performance of 100%, 98.5%, and 93.5% for large, medium, and small fish bones, respectively [162]. Calibration equations were obtained by the partial least squares regression technique for the computed tomography, predicting an RMSEPCV of 0.97% when LMP scanned only the ham and 0.9% when scanning both the ham and the lumbar region [105]. Table 3 summarizes the application of machine vision systems using different illumination and image acquisition methods on various meat.

### 4.2. Data Processing Method

After image acquisition, it is very important to pass some form of dimensionality reduction. Common data processing algorithms used in machine vision are support vector machines, regression models, and artificial neural networks (ANN).

#### 4.2.1. Support Vector Machine

Geronimo et al. used a machine vision system and spectral information from the near-infrared region to distinguish between wood breasts and normal chicken breasts. Combining image analysis with a support vector machine classification model, 91.8% of chicken breasts were correctly classified, while NIR showed only 97.5% accuracy [164]. Assia et al. implemented an embedded system based on DSP, PCA, and SVM algorithms using a dataset of 81 HSI beef images for projection and prediction models using PCA and SVM for beef classification and identification to detect the freshness of beef during refrigeration with a 100% success rate of recognition and classification [165]. Sun et al. developed a machine vision system for measuring and building an artificial intelligence prediction model for pork color and marbling quality grade. The prediction accuracy of the measured pork color using SVM was 92.5% and the prediction accuracy of the measured pork marbling was 75.0% [166]. Liu et al. (2018) investigated the ability of machine vision systems to predict the intramuscular fat percentage (IMF%) of pork. The correlation between subjective IMF% and ether extract IMF% was 0.81, while the correlation between image IMF% and ether extract IMF% was 0.66. The accuracy of the stepwise regression model was 0.63 and the support vector machine was 0.75 [167].

#### 4.2.2. Regression Models

Sun et al. developed a color-based machine vision system that uses digital image analysis to efficiently segment the adipose tissue in pork loin samples to predict the color properties of pork loin, using a linear regression model with a coefficient of determination of 0.83, a higher correlation than the stepwise regression model (r = 0.7) [168]. Chen et al. measured the actual intramuscular fat content (IMF%) and meat color of 200 pigs and then compared them with the scoring results of the machine vision system to construct an estimation model using SR and gradient boosting machine (GBM), with model accuracies of 0.875 and 0.89 for SR and GBM based on residual distribution [169]. Aass et al. used a procedure developed with a stepwise regression model to study the accuracy and precision of intramuscular fat prediction (USIMF) in lean cattle [157].

#### 4.2.3. Artificial Neural Networks

Huang et al. used integrated NIR, machine vision, and electronic nose techniques to measure TVB-N content in pork. PCA was used to fuse data from different sensor feature variables and a back propagation artificial neural network (BP-ANN) was used to construct a TVB-N content prediction model [149]. Huang et al. obtained image information of sensory changes and spectral information of structural changes in fish samples during storage, compressed and reduced the dimensionality of the data using PCA, and used a BP-ANN to build a prediction model of fish freshness and changes in fish during storage [150]. Huang et al. used a residual network (ResNet) to extract pork features to classify the types of pork original cuts into four categories: Ham, loin, belly, and neck, and applied machine vision to recognize different pork cuts with an accuracy of 94.47%. The main reasons that the different recognition results for some images still exist are the effects of the dataset size and the lighting environment in which the images were taken [170]. Table 4 summarizes the application of machine vision systems using different data processing methods on various meat.

## 5. Electronic Nose

The electronic nose provides the overall information and implied characteristics of the sample under test through various types of sensors and different pattern recognition systems [163].

The odor composition of food products is complex, and traditional odor analysis techniques are difficult to identify. On the other hand, sensory analysis by experts is expensive and subjective, thus the application of electronic noses is gradually becoming widespread [171,172]. Unlike chromatographs and spectrometers, the electronic nose is not a quantitative and qualitative analysis of one or several components of the sample under test, but a “fingerprint” of the volatile components in the sample.

The electronic nose essentially simulates the human olfactory organ for odor perception and analysis, which mainly consists of gas transmission sampling, sensor processing, signal pre-processing, pattern recognition, odor expression, and other units [173]. When a certain odor encounters a sensor corresponding to the active material, the chemical components in the odor will interact with the active material and the sensor can detect the transformation of the chemical signal into an electrical signal, which can only be processed with a suitable pattern recognition algorithm after appropriate pre-processing (noise elimination, feature extraction, signal amplification, etc.) [174]. The sensor is the most important part of the electronic nose, and its role is to collect information about the measured parameters [174]. The electronic nose has a short analysis time and does not damage the test sample, but the sensor technology should be highly improved for an electronic nose system [171].

### 5.1. Electrochemical Sensors

The most widely used electrochemical sensors are conductivity sensors: Metal oxide semiconductor (MOS) sensors, metal oxide semiconductor field effect transistor sensors (MOSFET), and conductive polymer sensors (CP).

#### 5.1.1. MOS Sensors

The selectivity and sensitivity of MOS sensors depend mainly on the semiconductor material used in the sensor [175]. The semiconductor material used in MOS sensors drifts with changes in ambient humidity and also reacts “poisonously” to sulfides present in gas mixtures, which tends to reduce the detection accuracy of the electronic nose.

An electronic nose consists of eight different MOS gas sensors for distinguishing mackerel, anchovy, and whiting, and a classification algorithm based on a binary tree structure could achieve an overall accuracy of 96.18% for fish recognition [176]. The aroma compounds in Siniperca chuatsi were analyzed using an electronic nose equipped with 10 different MOS sensors in combination with GC-MS [177]. An electronic nose consisting of 18 MOS gas sensors was studied to measure and simulate flavor quality changes in refined chicken fat during controlled oxidation [178]. An electronic nose was applied to detect adulteration of lamb combined with PLS, MLR, and BPNN, which served as a prediction model for the content of pork in minced lamb [179]. The use of an electronic nose containing 18 MOS sensors was studied to differentiate between chicken and beef seasonings and predict sensory attributes [180]. An electronic nose consisting of 10 MOS sensors could extract flavor fingerprints of Chinese-style sausages during processing and storage to assess lipid oxidation [181].

An electronic nose and an electronic tongue with six MOS sensors were designed to identify not only three different sources of red meat, but also the number of days of refrigerated storage successfully [182]. MOS sensor-based electronic nose was investigated to identify spoiled or contaminated fish and beef using ANN, SVM, and k-nearest neighbor for data analysis and comparison [183]. Olafsdottir et al. investigated the feasibility of a prototype gas sensor array system for fish volatile organic compounds (VOC), establishing quality criteria based on sensory attributes (sweet/sour, off-flavors, and putrefactive aromas) and classifying samples based on the response of an electronic fish nose [184]. Haugen et al. demonstrated the feasibility of direct quality measurements of smoked salmon with electronic noses, which were used to monitor quality changes and compare results with traditional sensory, chemical, and microbiological measurements [185]. Chantarachoti et al. evaluated the ability of a portable electronic nose to detect spoilage of whole Alaskan pink salmon stored at 14 °C and in slush ice. The instrument classified fish as fresh or spoiled with 92% accuracy [186].

#### 5.1.2. MOSFET

MOSFET has the low sensitivity to ammonia and carbon dioxide as well as baseline drift, which is the main source of error for electronic noses using MOSFET [187].

An electronic nose containing eight MOS sensors and six MOSFET sensors and gas chromatography-mass spectrometry were used to measure volatile compounds in meatballs and perform sensory analysis [188]. The electronic nose could clearly distinguish between spoiled broiler packages and fresh packages, both in the early stages and in the period of spoilage when sensory changes were evident, and the numbers of enterobacteriaceae and hydrogen sulfide-producing bacteria were most consistent with the electronic nose results, fully indicating that the electronic nose was able to detect early signs of spoilage in air-seasoned packaged poultry meat [189].

#### 5.1.3. Conductive Polymer Sensors

Conductive polymers (CP) have been used as active layers in gas sensors since the early 1980s [190]. Sensing polymer-based nanocomposites (CPNC) were effectively enhanced in terms of sensitivity and selectivity [191]. CP sensors require less electrical energy to operate at room temperature and respond quickly but are more sensitive to ambient relative humidity.

The ability of a CP sensor E-nose was evaluated to detect the presence of off-flavored malodorous compounds in catfish fillets [192]. An electronic nose with 32 CP sensors was used to determine the overall antioxidant status of fresh meat from animals fed different diets and to distinguish them by their odor characteristics [193].

### 5.2. Piezoelectric Sensors

Two types of piezoelectric sensors are commonly used in electronic noses, one is a body acoustic wave sensor (including quartz crystal microbalance sensors) and the other is a surface acoustic wave sensor (SAW) [174].

The volatile compounds were studied in meat products for halal certification using an electronic nose with a single uncoated quartz surface acoustic wave sensor and a gas chromatography-mass spectrometer with a headspace analyzer (GCMS-HS) [194]. The performance of an electronic nose based on a portable quartz microbalance was evaluated for monitoring spoilage of aerobically packed beef tenderloin at different storage temperatures [195].

While electronic noses offer many advantages over conventional analysis, the sensors also have some unresolved drawbacks. These include issues, such as sensor poisoning, sensor drift, and sensitivity. Selectivity and sensor drift has been the focus of research. Recent trends in overcoming sensor drawbacks include combining semiconductor chemical sensors with other types of sensors. While this complicates the sampling system (requiring more volume and electronics), this hybrid technology can compensate for the shortcomings of current chemical sensor technology. Hyperspectral imaging and electronic nose (E-nose) techniques were combined to improve the feasibility of freeze-thaw pork moisture (MC) prediction performance. The spectral and image (color and texture) information was extracted from the HSI sensor, while the odor information was extracted from the E-nose sensor to detect MC in pork. The method is an effective data fusion method, and the combination of HSI and E-nose techniques improves the prediction performance of MC in freeze-thawed pork [196]. Table 5 summarizes the application of different sensor-based electronic noses on some types of meat.

## 6. Discussion

Traditional meat detection techniques mainly include sensory evaluation, microbiological testing, and physicochemical experiments, etc. Sensory evaluation is quite subjective owing to its high dependence on the vision, smell, taste, and touch of the testing expert. It requires the processing temperature, processing time, and substrate for meat products to be parallel and operated according to the standard procedures during sample preparation. In addition, it demands to be equipped with multiple evaluators, which is a huge drain on human resources and time, and the current state of sensory evaluators also has an important impact on the final results. Microbiological testing and physical-chemical experiments are strict in terms of experimental environment and testing process, resulting in over-consumption of time and samples. In addition, irreversible damage will occur to the samples during the procedure of experiments.

Non-destructive testing is an emerging technology to effectively solve the limitations of traditional testing methods with the advantages of fast, accurate, and non-invasive detection. Nowadays, non-destructive detection technology (NDDT) has been widely used for freshness detection, adulteration identification, odor detection, and detection of certain compounds in meat products. It plays an important role in meat quality assurance and has become one of the most important tools for controlling the safety of edible meat. It shows clear advantages compared with traditional detection techniques and has a wide range of prospects in the field of meat detection.

Recently, with further improvements in detection instruments and data processing algorithms, non-destructive techniques make great contributions to the analysis of meat and meat products. However, it cannot be assumed that NDDT can completely replace traditional detection techniques. There is no doubt that color, flavor, and texture are important criteria to measure the quality and freshness of meat. Sensory evaluation, which highly relies on the sense of evaluation experts, seems to be more subjective than other detection methods. In fact, both NDDT and traditional detection techniques are devoted to ensuring the quality of meat to protect human health. However, the exact instruments are inconvenient to carry when humans visit the market and it will take a great deal of time to wait for the results, thus judging the freshness of meat by human senses is the most convenient and widely used method for them. In addition, people from different countries possess diverse demands on the quality of meat. For example, in China, people are in favor of pork and chicken, while in England, they tend to eat beef. NDDT is good at predicting and classifying the quality of meat according to defined standards. It cannot completely replace the traditional detection techniques and should be used in combination with them.

## 7. Challenges and Outlooks

Despite the rapid development of spectroscopic and imaging techniques, they are faced with a few drawbacks. Spectroscopic techniques have the ability of rapid analysis and high sensitivity and have been widely researched on the classification and shelf-life detection of meat products. Among them, NIR analysis is fast and non-destructive, providing a good prediction of the sensory properties of meat. It possesses the property of high penetrating power and the instruments are often simple and easy to maintain, with no need for tedious pretreatment and abundant samples. However, NIR analysis method requires a large number of physicochemical experiments before modeling and the model needs to be continuously updated according to the classes and states of samples. In addition, NIR spectroscopy has a low chance of non-resonant absorption leaps, which makes for its relatively low sensitivity, and it is not suitable for the analysis of aquatic organisms due to the disturbance of oxhydryl in water. Although NIR spectroscopy is in the leading position when applied for the detection of red meat, it is sensitive to external factors, such as ambient temperature during measurement [197]. In the future, continuous researches will appear, concentrating on extracting effective spectral information and improving the signal-to-noise ratio. Raman spectroscopy is a green detection technique with no need for sample pretreatment or preparation process, which has the advantages in terms of information generation. It requires only a small number of samples to obtain the key characteristics of compounds, avoiding the prediction errors to a certain extent. In truth, Raman spectroscopy is a rather weak phenomenon, which depends on the inelastic scattering of photons, and since the Raman effect is less intense than fluorescence, even a very small amount of fluorescence will cause the contamination of tested samples [198].

Imaging technique has a high adaptability to detect changes in chemicals in meat. Hyperspectral imaging is designed to enable efficient and reliable measurements of the content and spatial distribution of multiple chemical components and physical properties simultaneously [199]. It can determine the color and performs better than RGB imaging [87]. Although hyperspectral imaging has achieved great progress in the qualitative analysis of meat and meat products, the accuracy of outcomes is still not high. To solve this problem, it is desirable first to improve the precision of spectral devices and reduce the interference of useless information. In addition, the high dimensionality of hyperspectral data limits the processing speed of hyperspectral imaging technique in order that more efficient data processing algorithms need to be developed [8]. At present, HSI has been widely used in the field of detecting defective fruits and tea, and there will be a low cost and real-time meat detection system available for the modern industry in the future. Thermal imaging is a non-destructive and non-contact temperature sensor technology. It can capture the moving targets in real time and generate a visual image to show the difference in temperature over a wide range. This technology has the advantages of long range, high penetrating power, resistance to strong light interference, and adaptability to night and harsh conditions. It records the heat distribution by measuring the infrared radiation emitted from the surface of objects. Therefore, it is sensitive to the thermal interference in the environment, and the results may be uncertain when detecting targets with unstable temperatures. It is necessary to reduce the high cost and weaken the influence of thermal interference from the surrounding environment before it can be widely used in industry [154].

Machine vision is commonly fast, efficient, and comprehensive, and it tends to describe the spatial information of samples and cannot identify their internal characteristics. During the process of imaging, computer vision is strict with the external environment and lighting condition, especially when detecting the samples taken by cell phones and cameras. Moreover, the size of samples exercises considerable influence over the accuracy of results. It causes a small proportion when the samples are divided into tiny parts, and it is difficult to extract effective features when the samples are significantly large. Furthermore, the complexity of environment, such as the low contrast of color between the samples and background, will bring about serious noise interference. Machine vision is an effective technique for meat quality assessment and is most likely to be expanded to daily life. In the future, it is promising to equip cell phones with machine vision systems in order that consumers can quickly know the quality and freshness of meat when they select the products.

Electronic nose has the advantage of being objective, accurate, and low cost. It can detect the overall information of volatile substances and toxic components in the samples. In recent years, with the development of sensor technology, electronic nose has confronted many opportunities and challenges. For example, semiconductor materials will drift along with the change in environment, and MOS sensors produce a “toxic” response to sulfides. Furthermore, the CP sensor is sensitive to environmental humidity response, thus electronic noses need a system to prevent sensor drift [198]. The majority of electronic nose applications exhibit poor reproducibility and predictability without extensive calibration and mathematical analysis of the sensors. More importantly, sensor arrays and pattern recognition techniques tend to evaluate the quality of samples, which have been rarely applied for providing the data related to composition and concentration. The current researches on electronic nose mostly stay in the laboratory stage and have not been promoted for industrial environment. Nanomaterials, with the advantages of high sensitivity and selectivity, are expected to be applied to the electronic nose systems in the future.

In general, machine vision describes the external characteristics of meat, but it is difficult to obtain the internal quality of meat. NIR can detect the changes in the internal composition of meat while it is unsuitable for the identification of external information. With regard to electronic nose, it is mainly used to monitor the volatile gases released from the meat, and HSI can simultaneously predict the internal characteristics and external spatial information of samples [144]. In practical application, the quality of meat involves both external and internal factors, thus the integration of multiple inspection methods is a necessary development trend for future meat detection. Hyperspectral imaging was combined with machine learning to accurately detect the adulteration status of minced meat [200]. A combination of NIR spectroscopy, machine vision, and electronic nose was applied to measure the TVB-N content in pork, and the results were more advantageous than those with a single technique [149]. NIR spectroscopy and Raman spectroscopy are complementary when detecting the internal quality of meat, and their combination can improve the stability and accuracy of detection results. In addition, current studies mainly focus on red meat, and the detection of white meat tends to choose poultry and fish, rarely using amphibians (bullfrogs), crustaceans (crabs and shrimp), and bivalves (clams and oysters). In fact, the seafood occupies a significant proportion of meat and meat products, and it is a protein-rich sustainable food source. In the future, NDDT is hopeful to be expanded to the analysis of other categories of meat.

## 8. Conclusions

Meat is an important food for people to obtain protein and micronutrients daily. This paper provides an overview of traditional methods for testing meat quality and briefly describes the principles and applications of spectroscopic techniques, imaging techniques, machine vision, and electronic nose applied for meat quality and safety evaluation. The advantages and disadvantages of NDDT are summarized by comparison with the traditional detection techniques.

In general, traditional detection technology is related to consumers’ experiences and habits as well as life and culture factors. It is practically impossible for people to visit the market with all types of testing instruments, thus judging the quality of meat by human senses is the most convenient and widely used method. On the other hand, consumers’ needs for meat quality vary by living environment and age, which results in differences in the criteria for judging meat quality. However, NDDT tends to detect and differentiate meat quality according to defined standards and cannot be set independently according to the needs of different consumers, thus the relationship between NDDT and traditional inspection techniques is complementary.

Although a specific NDDT indeed plays a great role in meat quality and safety evaluation, there are still some limitations. In the future, combining a variety of NDDT is an inevitable trend for the detection of meat. More importantly, NDDT is not a complete replacement for traditional detection techniques, and connecting the traditional methods with new approaches will result in the greatest outcome. Furthermore, it is necessary to take into consideration that meat is always diverse and NDDT should be further expanded for the detection of other types of meat and meat products.

## Figures and Tables

**Table 1 foods-11-03713-t001:** Summaries of different spectroscopic techniques applied for meat detection.

Techniques	Samples	Applications	Methods	Reference
Near-infrared spectroscopy	Chicken	Classification of poultry samples	DT and SVM	[55]
	Pork	Prediction of fatty acid profile	SVM, DT, and REPTree	[56]
	Minced lamb	Detection of adulteration with duck meat in minced lamb	PLSR	[57]
	Chicken	Discrimination of fresh and freeze-thawed chicken meat	SIMCA and PCA	[58]
	Beef	Prediction of WBSF, L*, a*, and b*	PLSR	[59]
	Duck	Prediction of TVB-N content	PLSR and PCA	[60]
	Pork	Identification of repeatedly frozen meat	PCA and SCNN	[61]
	Pork	Evaluation of pH and color	PLSR	[62]
	Pork	Prediction of post-mortem meat quality (pH, drip loss, and intramuscular fat)	PLSR	[63]
	Duck	Evaluation of pH and color	PLSR	[64]
	Lamb	Classification of geographical origins and prediction of δ^13^C and δ^15^N	PCA, PLS-DA, LDA, and PLSR	[65]
	Lamb	Classification of geographical origins	PCA + LDA and PLS-DA	[66]
	Fishmeal	Discrimination of meat and bone meat in fishmeal	PLS-DA	[67]
	Crab	Determination of edible meat content	PLSR	[68]
	Sliced pork meat	Evaluation of freshness and detection of spoilage	PCA, CDA, and PLS	[69]
	Freeze-dried beef and mutton	Prediction of IMF and protein content	PCA and PLSR	[70]
	Heated fish and shellfish meats	Determination of end-point temperature	MLR	[71]
Raman spectroscopy	Beef, pork, and mutton	Identification of species	RF and BPNN	[72]
	Beef, lamb, venison	Classification of species	SVM and PLS-DA	[73]
	Beef	Assessment of tenderness	PLSR	[74]
	Beef	Prediction of WBSF, IMF, ultimate pH, drip-loss, and cook-loss	PLSR	[75]
	Dairy bull beef	Assessment of physico-chemical traits related to eating quality	PLSR	[76]
	Beef and poultry	Identification of meat-associated pathogens	SVM	[77]
	Beef	Detection of frauds in bovine meat by the addition of salts and carrageenan	PLS-DA	[78]
	Pork	Prediction of meat quality traits	PLSR	[79]
Terahertz spectroscopy	Pork	Prediction of freshness	BP-ANN and AdaBoost	[80]
	Chicken	Analysis of chlortetracycline hydrochloride and tetracycline hydrochloride	PLSR	[81]
	Fish, beef, chicken, and pork	Classification of species	PCA-SVM	[82]

Abbreviations: WBSF, Warner-Bratzler shear force; L*, lightness; a*, redness; b*, yellowness; TVB-N, total volatile basic nitrogen; IMF, intramuscular fat; DT, decision tree; RF, random forest; SVM, support vector machine; PLS, partial least squares; PLSR, partial least squares regression; PCA, principal component analysis; LDA, linear discriminant analysis; CDA, canonical discriminant analysis; PLS-DA, partial least squares discriminant analysis; MLR, multiple linear regression; SCNN, self-organizing competitive neural network; BPNN, back propagation neural network; BP-ANN, back propagation-artificial neural network; AdaBoost, adaptive boosting.

**Table 2 foods-11-03713-t002:** Summaries of different imaging techniques applied for meat detection.

Techniques	Samples	Applications	Methods	Reference
Hyperspectral imaging	Beef	Estimation of fat marbling	PLSR	[118]
	Pork	Prediction of fat content	SVR and PLSR	[119]
	Mutton	Discrimination and analysis of adulterated mutton	CNN	[120]
	Mutton	Evaluation of texture parameters	DT, RF, PLSR, and LSSVM	[121]
	Beef, lamb, and venison	Prediction of IMF and pH	PLSR and DCNN	[122]
	Lamb	Prediction of stearic acid content	PLSR and LSSVM	[123]
	Mutton	Detection of fatty acid content	SPA, UVE, and VCPA	[124]
	Chicken	Prediction of quality traits and grades of intact chicken breast fillets	PLSR	[125]
	Pork	Assessment of IMF quality	PLSR	[126]
	Beef, chicken, mutton	Classification of minced meat	SVM	[127]
	Chicken	Determination of freshness	RF and PLSR	[128]
	Pork	Identification of jowl meat adulteration in pork	PLSR	[129]
	Red meat	Discrimination of red meat	LDA, PLS-DA, and SVM	[130]
	Chicken	Detection of spoilage	Optimized BPNN	[131]
	Mutton and fox meat	Detection of fox meat adulteration in mutton	PLSR and SVR	[132]
	Beef and chicken	Visualization of the percentage of adulterated meat	LS	[133]
X-ray imaging	Beef	Prediction of composition	Linear mixed effects model	[134]
	Meat	Detection of needles in meat	CNN	[135]
	Pork	Prediction of softness	PLSR	[136]
	Lamb	Determination of the proportions of fat, lean, and bone	General linear model	[137]
	Lamb	Prediction of IMF content and SF	MLR	[138]
	Pork	Assessment of carcass composition	SA	[139]
	Pork	Prediction of lean content	PLSR	[140]
	Beef	Prediction of composition, fatty acids, and quality characteristics	PLSR	[141]
Thermal imaging	Pork and Mutton	Classification of minced mutton adulteration with pork	CNN	[114]
	Beef	Determination of temperature distributions	OT and CT	[142]
	Pork	Detection of meat quality defects	Infrared thermography	[143]

Abbreviations: SF, shear force; IMF, intramuscular fat; DT, decision tree; RF, random forest; SVM, support vector machine; SVR, support vector regression; LS, least square; LSSVM, least squares support vector machine; LDA, linear discriminant analysis; PLS-DA, partial least squares discriminant analysis; PLSR, partial least squares regression; CNN, convolutional neural network; DCNN, deep convolutional neural network; BPNN, back propagation neural network; SPA, successive projection algorithm; UVE, uninformative variable elimination; VCPA, variable combination cluster analysis; SA, statistical analysis; MLR, multiple linear regression; OT, ohmic thawing; CT, conventional thawing.

**Table 3 foods-11-03713-t003:** Summaries of different illumination and image acquisition methods in machine vision systems applied for meat detection.

Samples	Applications	Image Acquisition Methods	Lighting Methods	Reference
Pork	Detection of defects in pork loin longus muscle	Macro lens digital camera	Halogen lamp	[148]
Pork	Determination of TVB-N in pork	High-performance charge-coupled device (CCD) cameras	2 × 18 W lighting tubes produce light scattering	[149]
Fish	Development of a new rapid non-destructive technique to assess fish freshness	CCD camera with 8 mm zoom lens	Fluorescent and LED lamps	[150]
Beef, pork, chicken	Limits of colorimeter and image analysis techniques when evaluating the color of beef, pork, and chicken	Digital camera with CMOS sensor	4 fluorescent lamps and light diffusers	[151]
Fish	Assessment of the freshness of the fish	Color Digital Camera	4 fluorescent lamps	[152]
Tilapia	Development of a machine vision system based on pupil and gill color changes in tilapia	Color Digital Camera	4 fluorescent lamps provide diffuse reflection	[163]
Beef	Accurate estimation of beef marbling standards (BMS) quantities	Ultrasound Echo Signal	/	[155]
Beef	Estimate of the number of marbling in live cattle	Ultrasound systems with universal transducer arrays	/	[156]
Beef	The accuracy and precision of intramuscular fat prediction in lean cattle	Scanners with quality ultrasound indexing programs and transducers	/	[157]
Meat	Study of the distribution of several textural characteristics within meat products	Nuclear Magnetic Resonance Imaging	/	[158]
Meat	The structural and physical changes in meat during cooking	Horizontal 4.7T MRI system	/	[159]
Pork	Assessment of whether magnetic resonance imaging (MRI) can reliably analyze components in pigs	Nuclear Magnetic Resonance Imaging	/	[160]
Lamb	Development of a new muscle index for the hind leg and lumbar region of lambs	CT	/	[161]
Salmon, trout	Automatic detection of fish bones in fish fillets	CT	/	[162]
Meat	Prediction of carcass leanness ratio and different cuts	CT	/	[105]

Abbreviations: TVB-N, total volatile base nitrogen; CCD, charge-coupled device; LED, light emitting diode; CMOS, complementary metal oxide semiconductor; BMS, beef marbling standards; MRI, magnetic resonance imaging; CT, computed tomography.

**Table 4 foods-11-03713-t004:** Summaries of different data processing methods in machine vision systems applied for meat detection.

Techniques	Samples	Applications	Methods	Reference
CV	Pork	Measure and build artificial intelligence prediction models for pork color and marbling quality grades	SVM	[166]
	Pork	Prediction of percentage of intramuscular fat in pork (IMF%)	SVM	[167]
	Pork	Prediction of the color properties of pork tenderloin	Linear Regression, Stepwise Regression	[168]
	Beef	The accuracy and precision of intramuscular fat prediction in lean cattle	Stepwise Regression	[157]
	Pork	Identification of the different cuts of pork	ResNet	[170]
CV, NIR, E-nose	Pork	Determination of TVB-N in pork	PCA, BP-ANN	[149]
CV, NIR	Fish	Development of a new rapid non-destructive technique to assess fish freshness	PCA, BP-ANN	[150]
	Chicken	Distinguishment of the woody and normal chicken breasts	SVM	[164]
CV, DSP	Beef	Classification of the freshness of beef	PCA, SVM	[165]
CV, Traditional Method	Pork	Estimation of the intramuscular fat content of pork	Linear Regression, Stepwise Regression, GBM	[169]

Abbreviations: TVB-N, total volatile base nitrogen; PCA, principal component analysis; BP-ANN, back propagation artificial neural network; CV, computer vision; NIR, near-infrared; E-nose, electronic nose; SVM, support vector machine; DSP, digital signal processing; IMF, intramuscular fat; GBM, gradient boosting machine; ResNet, residual neural network.

**Table 5 foods-11-03713-t005:** Summaries of different sensors in electronic nose systems applied for meat detection.

Techniques	Samples	Applications	Methods	Sensor Systems	Reference
E-nose	Fish	Discrimination of fish species	New Approach, Naïve Bayes, k-Nearest Neighbor, LDA	8 different MOS sensors with temperature and humidity sensors	[176]
	Chicken	Measurement and simulation of flavor quality changes in refined chicken fat during controlled oxidation	PLSR	18 MOS gas sensors	[178]
	Chicken and beef	Distinguishment of chicken and beef seasonings	PCA, PLSR	18 MOS sensors	[180]
	Chinese- style sausage	Extraction of flavor fingerprints of Chinese-style sausages during processing and storage	SVM, ANN	10 MOS sensors	[181]
	Beef and fish	Identification of spoiled or contaminated fish and beef	ANN, SVM, k-nearest neighbor	MOS sensor	[183]
	Fish	Classification of fish according to sensory attributes	PCA, PLSR	MOS gas sensor arrays	[184]
	Salmon	Evaluation of the feasibility of direct quality measurements on salmon and detected changes in salmon quality	PCA, PLSR	MOS sensor	[185]
	Pink salmon	Assessment of the ability of electronic noses to detect pink salmon spoilage at 14 °C	PCA	MOS sensor	[186]
	Chicken	Study of the applicability of electronic noses for the quality control of modified atmosphere packaged broiler chicken cuts	PCA, PLS, ANN	12 MOS sensors and 10 MOSFET sensors	[189]
	Catfish	Detection of malodorous compounds in catfish fillets and assessment of the potential marketability of the meat	PCA	CP sensor	[192]
	Beef	Determination of the overall antioxidant status of fresh meat from animals fed with different diets	PCA, LDA	32 CP sensors	[193]
	Beef	Evaluation of the performance of monitoring aerobically packed beef tenderloin spoilage at different storage temperatures	SVM, Regression Model	Portable QMB sensor	[195]
E-nose, GC-MS	Siniperca chuatsi	Analysis and comparison of aroma compounds of Siniperca chuatsi during fermentation and storage	PCA	10 different MOS sensors	[177]
	Meatball	Measurement of volatile compounds in meatballs and sensory analysis	PLSR	8 MOS sensors and 6 MOSFET sensors	[188]
	Pork	Study of volatile compounds in meat products and halal certification	PCA	Single uncoated quartz SAW sensor	[194]
Traditional Methods, E-nose	Pork	Prediction of pork content in lamb	PLS, MLR, BPNN	MOS sensor	[179]
E-nose, E-Tongue	Red Meat	Identification of the different sources and days of refrigeration of red meat	PCA, SVM	6 MOS sensors	[182]
E-nose HSI	Pork	Prediction of moisture content (MC) in frozen-thawed pork	PLSR	/	[196]

Abbreviations: MOS, metal oxide semiconductor; LDA, linear discriminant analysis; E-nose, electronic nose; PCA, principal component analysis; GC-MS, gas chromatography-mass spectrometry; PLSR, partial least square regression; PLS, partial least squares; MLR, multiple linear regression; BP-ANN, back propagation artificial neural network; ANN, artificial neural network; MOSFET, metal oxide semiconductor field effect transistor; CP, conductive polymer; SAW, surface acoustic wave; QMB, quartz crystal microbalance; SVM, support vector machine; HSI, hyperspectral imaging.

## Data Availability

Not applicable.

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
