# Peer review of "Non-Destructive Techniques for the Analysis and Evaluation of Meat Quality and Safety: A Review"

_foods, 2022, doi:10.3390/foods11223713_

Round 1
Reviewer 1 Report
General Comments:
I have reviewed the revision of manuscript titled “Non-destructive Techniques for the Analysis and Evaluation of Meat Quality and Safety: A Review”.
In the present study, the authors survey the different types of non-destructive detection techniques for the analysis and evaluation quality and safety of meat. The present study pointing that some of these NDDT are better to evaluate chemical composition, and some of them are better to analyse the nutritional value or physical quality of meat. The review clearly shows that non-destructive detection techniques have the advantages of fast, accurate, and non-invasive. On the other hand, it can be mentioned what the disadvantages of NDDTs are. Although the sensory evaluation (odour intensity, tenderness, juiciness, flavour intensity, flavour quality, and acceptability) of cooked meat is subjective, it varies according to the previous experiences and habits of the consumers. On the other hand, the consumption habits of consumers also change according to the society and culture in which they live. Considering this information, do the authors think that NDD techniques can be an alternative to sensory panels?
There is no doubt that various non-destructive detection techniques will become a mainstream trend in the future to achieve comprehensive analysis and assessment of meat quality and safety. In this respect, topic of the current study has scientific novelty and interest. The manuscript can be accepted as presented.
Title: It is considering that the title of the manuscript is representing accurately the contents of the review.
Abstract: Abstract concisely explains the present study.
Keywords: The keywords are suitable and appropriate.
Introduction: The purpose of the review is appropriately stated in the introduction part of the manuscript. The aim of the study is well understood.
Conclusion: It should be mentioned more widely what the disadvantages of NDDTs are.
Tables: The tables are sufficient and appropriate in the present manuscript.
Reviewer 2 Report
1.The outline represented by the title sounds feasible. However, although a lot of studies and a consistent common thread are rolled out by headings and subheadings, the preparation of the manuscript appears to be conducted with a lack of the necessary care. The manuscript is quite general.
2.Most importantly, cited references (sometimes) do not refer to the statements with which they are associated. I would like to ask the authors kindly to recheck all references and the assignment to certain statements carefully.
Other elements that I suggest to revise to improve the impact of the review:
3.The title is only reflected by the generally in the manuscript. If the focus of the review should be as suggested by the title, the aspect "prospects" may be enhanced e.g. discussing further strategies in more detail or by presenting studies that try to manipulate/optimize/counteract factors engaged in sea-food processing.
4.Table 1 is not standard, and due to a large amount of content, it is unreadable in the draft format of the manuscript (PDF file). Please re-write this table.
5. Description of material and methods are necessary - what was the way of searching references.
6. For the review papers, conclusion part is very critical so needs to address. Otherwise, there is an important missing part in the review. Furthermore, the authors did not add new perspectives on these researches and this creates a weakness. They should evaluate the conclusion on new approaches instead of known expectations as mentioned in several other papers.
Reviewer 3 Report
The manuscript presents different non-destructive techniques for the quality analysis of meat products. All techniques were clearly and concisely described presenting their advantages and disadvantages.
I believe that the manuscript did not present an innovative topic but I consider that the information is relevant as other alternatives to analyze the quality of meat products.
I suggest that the authors write about the advantages of these techniques compared to a conventional technique used in the quality analysis of meat products.
References are appropriate for the manuscript, although I suggest adding more information (two lines) about the work of the cited authors.
Regarding the tables, use the same format for all.
Round 2
Reviewer 2 Report
Manuscript can be accepted